# Risk factors for graft loss and death among kidney transplant recipients: A competing risk analysis

Jessica Pinto-Ramirez[1]*, Andrea Garcia-Lopez[2], Sergio Salcedo-Herrera[1], Nasly Patino-Jaramillo[2], Juan Garcia-Lopez[3], Jefferson Barbosa-Salinas[3], Sergio Riveros-Enriquez[3], Gilma Hernandez-Herrera[4], Fernando Giron-Luque[5]

1 Department of Transplant Nephrology, Colombiana de Trasplantes, Bogotá, Colombia, 2 Department of Transplant Research, Colombiana de Trasplantes, Bogotá, Colombia, 3 Departmento of Technology and Informatics, Colombiana de Trasplantes, Bogotá, Colombia, 4 Postgraduate Program in Epidemiology, Universidad del Rosario – Universidad CES, Bogotá-Medellín, Colombia, 5 Department of Transplant Surgery, Colombiana de Trasplantes, Bogotá, Colombia

* jpinto@colombianadetrasplantes.com

**Data Availability Statement:** All relevant data are available on Mendeley: http://dx.doi.org/10.17632/prrjh2f7xf.1.

## Abstract

### Introduction

Kidney transplantation is the best therapeutical option for CKD patients. Graft loss risk factors are usually estimated with the cox method. Competing risk analysis could be useful to determine the impact of different events affecting graft survival, the occurrence of an outcome of interest can be precluded by another. We aimed to determine the risk factors for graft loss in the presence of mortality as a competing event.

### Methods

A retrospective cohort of 1454 kidney transplant recipients who were transplanted between July 1, 2008, to May 31, 2019, in Colombiana de Trasplantes, were analyzed to determine risk factors of graft loss and mortality at 5 years post-transplantation. Kidney and patient survival probabilities were estimated by the competing risk analysis. The Fine and Gray method was used to fit a multivariable model for each outcome. Three variable selection methods were compared, and the bootstrapping technique was used for internal validation as split method for resample. The performance of the final model was assessed calculating the prediction error, brier score, c-index and calibration plot.

### Results

Graft loss occurred in 169 patients (11.6%) and death in 137 (9.4%). Cumulative incidence for graft loss and death was 15.8% and 13.8% respectively. In a multivariable analysis, we found that BKV nephropathy, serum creatinine and increased number of renal biopsies were significant risk factors for graft loss. On the other hand, recipient age, acute cellular rejection, CMV disease were risk factors for death, and recipients with living donor had better survival compared to deceased-donor transplant and coronary stent. The c-index were 0.6 and 0.72 for graft loss and death model respectively.

**Funding:** The authors received no specific funding for this work.

**Competing interests:** The authors have declared that no competing interests exist.

## Conclusion

We developed two prediction models for graft loss and death 5 years post-transplantation by a unique transplant program in Colombia. Using a competing risk multivariable analysis, we were able to identify 3 significant risk factors for graft loss and 5 significant risk factors for death. This contributes to have a better understanding of risk factors for graft loss in a Latin-American population. The predictive performance of the models was mild.

## Introduction

Kidney transplantation is the optimal renal replacement therapy for suitable patients with end stage renal failure [1]. Identifying risk factors for graft failure in kidney transplant recipients is useful for recognizing those patients at high risk and anticipating potential therapeutic interventions to improve graft survival [2, 3]. Risk factors in the field of organ transplantation are typically assessed using time-to-event outcomes, for instance, when recording time-to-death or time-to-graft loss. Survival analyses are key statistical tools in transplantation research [4]. This analyses are the most used methods to estimate the incidence of an outcome of interest, often censoring for a competing event [4]. For example, death is competing event to graft loss because a patient may die before losing the graft, as such no opportunity for graft loss in that case. Thus, competing events are present when another event precludes the event of interest. In this condition, the Kaplan-Meier (KM) approach is not suitable because this method assumes that censored patients are at the same risk as patients who remain in the study. In general, this leads to an overestimation of the cumulative incidence of the event of interest [5–7]. To solve this limitation, Kalbfleisch and Prentice introduced the Cumulative Incidence Function (CIF) [8]. The CIF calculates all events' probability as the sum of the event of interest's probabilities and those of the competing risks. The competing risk analysis (CRA) allows using a modified chi-squared test to compare CIF curves between groups and the Fine and Gray model with subdistribution hazards ($_{sd}HR$) [9, 10]. Thus, patients are followed until observing the first of multiple event types in the CRA. Adjusting this fact, the inferred incidence of the event of interest is lower, and the sum of calculated incidences across all event types sums up to 100% [4, 10].

The CRA may provide further insights into the effect of interventions on the separate endpoints, comparing CIF curves to explore the association between covariates and the absolute risk. Indeed, CRA may be essential for clinical decision-making and prognostic research questions [11]. Despite this advantage, competing risk models have not been used frequently by researchers [2, 10]. In particular, the advantage of using CRA method was highlighted in a study evaluating race, age, and survival among patients undergoing dialysis, where accounting for transplant as competing risk brought to light a greater disparity in death on dialysis among younger black patients (related to disparity in access to transplantation) [12]. Other studies used the CRA method to evaluate risk of mortality and subsequent graft survival in older recipients after sustaining fracture [13], as well as the risk of graft loss and mortality in older recipients (age ≥60 y) receiving older kidneys (age ≥80 y) versus remaining on dialysis [14]. So far, there are no Latin-American studies related to competing risks analysis in transplantation. In this context and given the substantial variability of the identified risk factors for graft loss across different transplant populations, our transplant program (Colombiana de Trasplantes—CT) aims to use a well characterized Latin-American cohort of kidney transplant recipients

with long term follow-up to determine the risk factors for graft loss in the presence of mortality as a competing risk event.

## Materials and methods

### Study design and population

We conducted a retrospective cohort study at Colombiana de Trasplantes. To give a context it must be said that Colombiana de Trasplantes is a transplant network in Colombia with 4 centers with around 21% of the annual national kidney transplant activity. We included first time, kidney transplant recipients aged $\geq$ 18 years who were transplanted between July 1, 2008, to May 31, 2019. Patients with primary renal graft thrombosis (arterial or venous) were excluded. Recipients were followed up to graft failure, death, or end of follow up at 5 years post transplantation, whichever was earliest.

### Kidney donors

Informed consent was obtained from organ/tissue deceased donors (DD) by a family interview where both family and the donation team go through information related to answer inquiries about encephalic death, emotional support, and the possibility of organ donation. The main causes of death in our DD were cerebrovascular/stroke, followed by head trauma, anoxia, CNS tumor, or others. Organ donation and tissue consent form is provided in S1 Table in S1 File.

In the case of the live kidney donors, our transplant team provides kidney donation and nephrectomy informed consent and the affidavit from the live kidney donors. Overall, less than 1% of our kidney donors were previously registered as organ donors. According to Colombian law, the total of donor medical costs is covered including organ donor maintenance and procurement with an average of 5000 USD and without any economic contribution to the family donor.

### Immunosuppression and follow-up protocol

All patients received standard induction therapy with alemtuzumab, basiliximab or antihuman thymocyte immunoglobulin according to immunologic risk or transplant clinical guidelines. All patients received a fixed-dose of methylprednisolone perioperatively for 3 days with a transition to fixed-dose oral prednisone from day 4 to day 7 in the postoperative period. One-week post transplantation steroids were withdrawn. Chronic immunosuppression consists of Calcineurin inhibitors-based therapy and mycophenolate mofetil. Patients are monitored closely in the first 4 weeks post transplantation, and they return for follow-up monthly thereafter.

The acute rejection was classified under parameters described by Banff (2015) [15]. Biopsy was performed on those patients with increase of serum creatinine by >20% from baseline. Our center does not perform biopsies per protocol.

Treatment for acute cellular rejection was started once the histological diagnosis was confirmed as follows: Methylprednisolone: 500 mg. IV / day in infusion for three days. Oral prednisolone from the fourth day at a dose of 0.5 mg / Kg / day divided into two doses and for two weeks. After completing the two weeks, a weekly decrease of 10 mg / day was made until reaching the previous dose that was received before the rejection episode. Serum creatinine was done 5 days after finishing the boluses. A response to corticosteroid treatment was defined with a decrease in serum creatinine greater than and equal to 20% of the patient's baseline creatinine.

Histocompatibility tests performed in our center correspond to Human Leukocyte Antigens (HLA), Panel Reactive Antibodies (PRA) I and II, flow cytometric crossmatch and anti-

HLA antibodies (the latter only in living donors when the crossmatch is positive). HLA matching is when the recipient and the donor shared the same HLA antigens (HLA-A,-B,-DR antigens) [16].

We do not perform routine preimplant biopsies. The decision to take or not the organs from expanded criteria donors is made by macroscopic evaluation of the graft and, if required, it is sent for histological evaluation. Indices like KDPI / KDRI are not considered for taking or allocating organs [17]. Organ allocation is made according to the allocation criteria for kidney transplantation in Colombia [18].

## Outcomes

Our primary outcome of interest was graft loss, not including death with function. Graft loss was defined by center reported as permanent return to dialysis or retransplantation. Death was defined as mortality from any cause and was ascertained by review of the Colombiana de Trasplantes database which records patient's death and supplemented with the National register Master File. Patients were censored at 5 years of follow-up since the last follow-up date if they were transferred to another transplant center or lost to follow-up. Thus, survival analyses were performed using a competing risk approach, where graft loss and mortality were treated as competing events.

## Statistical analysis

**Descriptive analysis.** Descriptive statistics were used to report the population characteristics. Frequencies and percentages were used for categorical variables. The numerical variables were reported according to its distribution using mean and standard deviation for normally distributed variables, and median and inter-quartile range (IQR) for non-normally distributed variables. Multiple imputation was not considered as there were few missing values (5.9% of the total number of cases), and those values were at random. According to this, we performed a complete case analysis in the univariable and multivariable models.

**Predictors.** Prespecified variables based on published literature and those available in our data, were collected as potential risk factors for graft loss. Data collected included demographics, medical history and clinic characteristics of kidney transplant recipient and donor. Definition of predictor measurement is provided in Supplementary material (S1 Table in S1 File).

**Incidence estimates.** The overall incidence of graft loss and/or death at 5 years post transplantation was calculated by Competing risk analysis method (CRA) using cumulative incidence function (CIF) where mortality was treated as a competing risk with graft loss. Log Rank test for graft loss and death were compared in the entire population and in specific patient population including living and deceased donor.

Comparisons between the two groups (graft loss yes/no and death yes/no) were performed using modified $\chi^2$ test. The subdistribution Hazard Ratio ($_{sd}$HR) also known as Fine and Gray model was calculated for each independent variable and the two outcomes.

**Variable selection and prediction.** Variables with p value <0.25 in an univariable analysis and those with clinical importance were selected to perform further analysis. Variable selection to build the final model for graft loss was performed comparing three methods:

1. Full model: contains all the predictors selected in previous analysis and no variable selection was done.

2. Backward selection based on the Akaike information criterion (AIC).

3. Backward selection based on the Bayesian Information Criterion (BIC).

The model was selected on the model´s better performance.

Bootstrapping technique was used for internal validation as split method for resample a single dataset to create many simulated samples. The prediction models were trained on B bootstrap samples with replacement. The models were assessed in the observations that were not included in the bootstrap sample. This allowed us to calculate the prediction error, brier score and c-index. Calibration plot was used to compare the predicted probability with the observed probability.

The Fine and Gray model directly models the covariate effect on CIF, and it reports the $_{sd}$HR. To model the impact of covariates on graft loss, we used the Fine and Gray method [9] for performing competing risk regression. The association between the primary outcome and the independent variables were assessed by the $_{sd}$HR.

The model development and report was based on The Transparent Reporting of a multivariable prediction model for Individual Prognosis Or Diagnosis (TRIPOD) [19]. More details of modeling process can be found in Supplementary material (Modelling process in S1 File).

Analysis was performed using the Software R version 3.6.3. Library used to perform competing risk analysis was *cmprsk* [20].

## Ethics considerations

This study was approved by the ethics research committee of the institution, acting in concordance with local and national regulations, as well as with the Helsinki declaration. Confidentiality of all patients was secured all the time during the execution of the research. None of the transplant donors was from a vulnerable population and all donors or next of kin provided written informed consent that was freely given.

## Results

### Patient characteristics

A total of 1454 out of 1621 recipients met selection criteria. Exclusions took place in 167 (113 pediatric transplants and 54 kidney transplants with graft thrombosis). In gender distribution most of patients were male, the overall mean age was 43.58 ± 13 years. Table 1 summarizes the clinical and demographic characteristics.

### Overall cumulative incidence

During the follow-up period graft loss occurred in 169 patients (11.6%) and death occurred in 137 (9.4%). Cumulative incidence for graft loss and death was 15.8% and 13.8% respectively. Fig 1 displays the combined cumulative incidence for the entire cohort. Significant differences in estimates of both outcomes were found when analyzing live and deceased transplant separately, where deceased transplant (17.1% and 16.3% for death and graft loss respectively) had greater cumulative incidences (deceased transplant 17.1% and 16.3% for death and graft loss respectively vs live 5.4% and 15% for death and graft loss respectively). Fig 2 shows the difference between type of transplant in the cumulative incidence of graft loss and death.

### Risk factors for cumulative incidence of graft loss and death with functioning graft

We fit the Fine and Gray competing risk survival regression model for identifying the potential determinants of graft loss using covariates with significant association and those with clinical importance. The covariates that had a significant impact on the graft loss were stroke, cold ischemia time, qualitative PRA II, BKV nephropathy, number of allograft biopsies, acute

**Table 1. Clinical and demographic characteristics of kidney transplant recipients.**

| Variable | Total | Graft loss | | P-value | Death | | P-value |
|---|---|---|---|---|---|---|---|
| | | No | Yes | | No | Yes | |
| | (N = 1454) | (N = 1285) | (N = 169) | | (N = 1317) | (N = 137) | |
| **Sex, n (%)** | | | | 0.495 | | | 0.323 |
| **Female** | 586 (40.3) | 525 (40.9) | 61 (36.1) | | 539 (40.9) | 47 (34.3) | |
| **Male** | 868 (59.7) | 760 (59.1) | 108 (63.9) | | 778 (59.1) | 90 (65.7) | |
| **Age, mean (SD)** | 43.6 (13.2) | 43.6 (13.1) | 43.2 (14.3) | 0.926 | 42.8 (13.2) | 50.9 (11.3) | <0.001 |
| **BMI, mean (SD)** | 23.3 (3.82) | 23.3 (3.79) | 23.2 (4.04) | 0.988 | 23.2 (3.81) | 24.1 (3.78) | 0.030 |
| Missing | 49 (3.4) | 39 (3.0) | 10 (5.9) | | 47 (3.6) | 2 (1.5) | |
| **Cause of CKD, n (%)** | | | | 0.946 | | | 0.076 |
| Congenital | 96 (6.6) | 84 (6.5) | 12 (7.1) | | 90 (6.8) | 6 (4.4) | |
| Unknown | 638 (43.9) | 569 (44.3) | 69 (40.8) | | 583 (44.3) | 55 (40.1) | |
| Diabetic | 200 (13.8) | 174 (13.5) | 26 (15.4) | | 167 (12.7) | 33 (24.1) | |
| Glomerular | 272 (18.7) | 241 (18.8) | 31 (18.3) | | 256 (19.4) | 16 (11.7) | |
| Hypertensive | 163 (11.2) | 145 (11.3) | 18 (10.7) | | 144 (10.9) | 19 (13.9) | |
| Obstructive | 37 (2.5) | 34 (2.6) | 3 (1.8) | | 35 (2.7) | 2 (1.5) | |
| Other | 48 (3.3) | 38 (3.0) | 10 (5.9) | | 42 (3.2) | 6 (4.4) | |
| **RRT, n (%)** | | | | 0.266 | | | 0.213 |
| Hemodialysis | 618 (42.5) | 531 (41.3) | 87 (51.5) | | 554 (42.1) | 64 (46.7) | |
| Peritoneal | 447 (30.7) | 406 (31.6) | 41 (24.3) | | 415 (31.5) | 32 (23.4) | |
| Pre-Dialysis | 181 (12.4) | 165 (12.8) | 16 (9.5) | | 168 (12.8) | 13 (9.5) | |
| Unknown | 208 (14.3) | 183 (14.2) | 25 (14.8) | | 180 (13.7) | 28 (20.4) | |
| **Time on dialysis, months (SD)** | 27.2 (35.4) | 26.7 (34.7) | 30.8 (40.4) | 0.421 | 26.6 (35.5) | 33.5 (34.2) | 0.149 |
| **Time on waiting list, months (SD)** | 554 (596) | 561 (602) | 484 (541) | 0.423 | 547 (594) | 615 (623) | 0.510 |
| **Medical history n (%)** | | | | | | | |
| CVD | 47 (3.2) | 38 (3.0) | 9 (5.3) | 0.262 | 41 (3.1) | 6 (4.4) | 0.728 |
| **Stroke** | 13 (0.9) | 8 (0.6) | 5 (3.0) | 0.010 | 12 (0.9) | 1 (0.7) | 0.977 |
| Hypertension | 1162 (79.9) | 1033 (80.4) | 129 (76.3) | 0.465 | 1045 (79.3) | 117 (85.4) | 0.242 |
| DM | 205 (14.1) | 179 (13.9) | 26 (15.4) | | 171 (13.0) | 34 (24.8) | <0.001 |
| Smoking | 210 (14.4) | 184 (14.3) | 26 (15.4) | 0.934 | 186 (14.1) | 24 (17.5) | 0.561 |
| **Type of donor, n (%)** | | | | 0.176 | | | <0.001 |
| DD | 1002 (68.9) | 875 (68.1) | 127 (75.1) | | 881 (66.9) | 121 (88.3) | |
| LD | 452 (31.1) | 410 (31.9) | 42 (24.9) | | 436 (33.1) | 16 (11.7) | |
| **ECD, n (%)** | 189 (13.0) | 158 (12.3) | 31 (18.3) | 0.157 | 155 (11.8) | 34 (24.8) | <0.001 |
| **CIT, hours mean (SD)** | 18.3 (14.4) | 17.8 (13.3) | 21.3 (20.3) | 0.039 | 18.3 (15.1) | 18.3 (7.74) | 1 |
| **CIT >14 hours, n (%)** | 675 (46.4) | 573 (44.6) | 102 (60.4) | <0.001 | 588 (44.6) | 87 (63.5) | <0.001 |
| **Match, n (%)** | | | | 0.808 | | | 0.628 |
| 0 | 117 (8.0) | 105 (8.2) | 12 (7.1) | | 110 (8.4) | 7 (5.1) | |
| 1 | 202 (13.9) | 175 (13.6) | 27 (16.0) | | 183 (13.9) | 19 (13.9) | |
| 2 | 298 (20.5) | 258 (20.1) | 40 (23.7) | | 267 (20.3) | 31 (22.6) | |
| 3 | 498 (34.3) | 447 (34.8) | 51 (30.2) | | 457 (34.7) | 41 (29.9) | |
| 4 | 220 (15.1) | 194 (15.1) | 26 (15.4) | | 189 (14.4) | 31 (22.6) | |
| 5 | 69 (4.7) | 60 (4.7) | 9 (5.3) | | 62 (4.7) | 7 (5.1) | |
| 6 | 40 (2.8) | 39 (3.0) | 1 (0.6) | | 39 (3.0) | 1 (0.7) | |
| Missing | 10 (0.7) | 7 (0.5) | 3 (1.8) | | 10 (0.8) | 0 (0) | |
| **Qualitative PRA I, n (%)** | | | | 0.321 | | | 0.477 |
| Negative | 748 (51.4) | 663 (51.6) | 85 (50.3) | | 685 (52.0) | 63 (46.0) | |
| Positive | 78 (5.4) | 63 (4.9) | 15 (8.9) | | 73 (5.5) | 5 (3.6) | |

*(Continued)*

**Table 1.** (Continued)

| Variable | Total | Graft loss | | P-value | Death | | P-value |
|---|---|---|---|---|---|---|---|
| | | No | Yes | | No | Yes | |
| | (N = 1454) | (N = 1285) | (N = 169) | | (N = 1317) | (N = 137) | |
| Unknown | 628 (43.2) | 559 (43.5) | 69 (40.8) | | 559 (42.4) | 69 (50.4) | |
| **Qualitative PRA II, n (%)** | | | | <0.001 | | | 0.859 |
| Negative | 95 (6.5) | 69 (5.4) | 26 (15.4) | | 87 (6.6) | 8 (5.8) | |
| Positive | 736 (50.6) | 662 (51.5) | 74 (43.8) | | 672 (51.0) | 64 (46.7) | |
| Unknown | 623 (42.8) | 554 (43.1) | 69 (40.8) | | 558 (42.4) | 65 (47.4) | |
| **CMV Disease, n (%)** | 76 (5.2) | 62 (4.8) | 14 (8.3) | 0.165 | 58 (4.4) | 18 (13.1) | <0.001 |
| **BKV nephropathy, n (%)** | 36 (2.5) | 21 (1.6) | 15 (8.9) | <0.001 | 35 (2.7) | 1 (0.7) | 0.385 |
| **Number of renal allograft biopsies, n (%)** | 1.01 (1.30) | 0.874 (1.20) | 2.08 (1.54) | <0.001 | 0.995 (1.30) | 1.20 (1.36) | 0.201 |
| **Acute cellular rejection, n (%)** | 473 (32.5) | 368 (28.6) | 105 (62.1) | <0.001 | 413 (31.4) | 60 (43.8) | 0.012 |
| **Serum creatinine at 12 months, mean (SD)** | 1.51 (0.848) | 1.42 (0.588) | 2.73 (2.10) | <0.001 | 1.48 (0.851) | 1.84 (0.748) | 0.004 |
| **Coronary stent** | 13 (0.9) | 12 (0.9) | 1 (0.6) | 0.906 | 9 (0.7) | 4 (2.9) | 0.030 |
| **Number of hospital readmissions, mean (SD)** | 1.30 (1.81) | 1.20 (1.77) | 2.07 (1.92) | <0.001 | 1.26 (1.79) | 1.70 (2.02) | 0.024 |

SD: standard deviation, BMI: Body Mass Index, RRT: Renal Replacement Therapy, CVD: Cardiovascular Disease, DM: Diabetes Mellitus. DD: Deceased donor, LD: Live donor, ECD: Expanded criteria donor, CIT: Cold isquemia time, PRA: Panel Reactive Antibody Test, BMI: Body Mass Index, CMV: citomegalovirus, BKV: BK virus. Negative PRA test is indicative of a lack of anti-HLA antibodies.

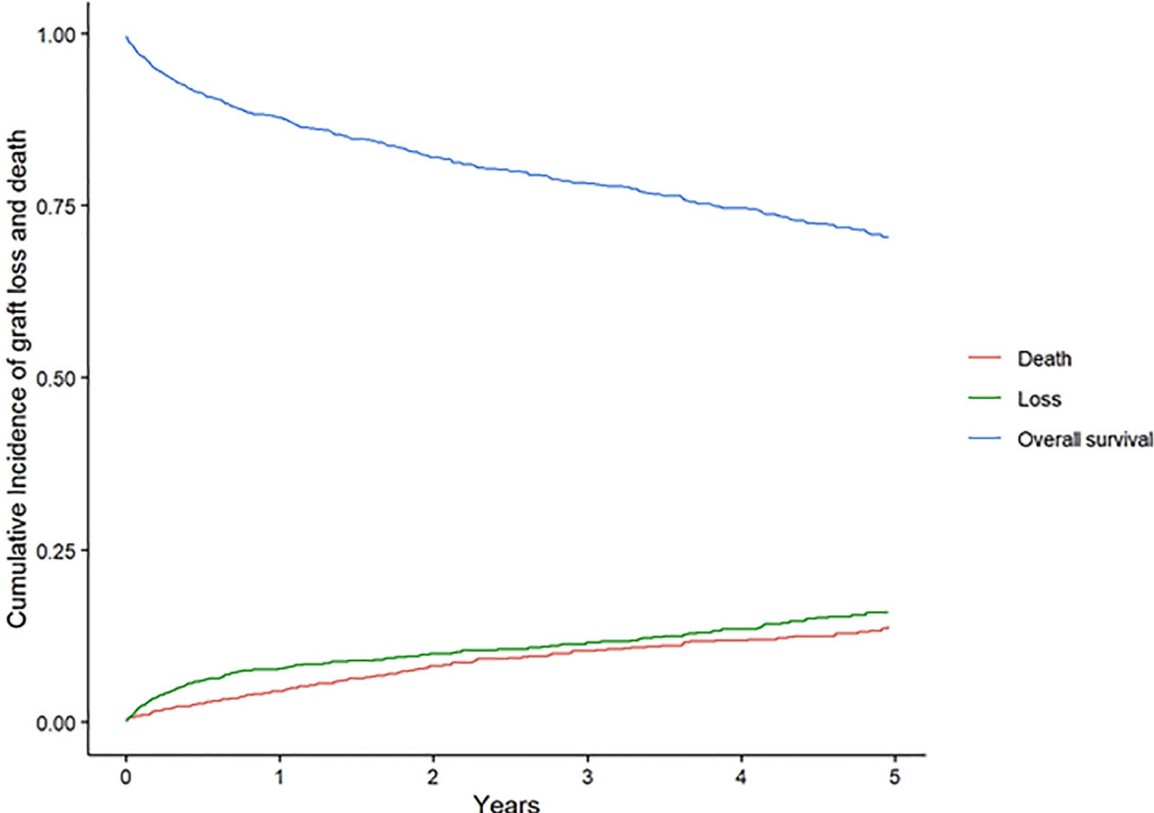

**Fig 1. Cumulative incidence of graft loss and death estimated by the method for competing risk.**

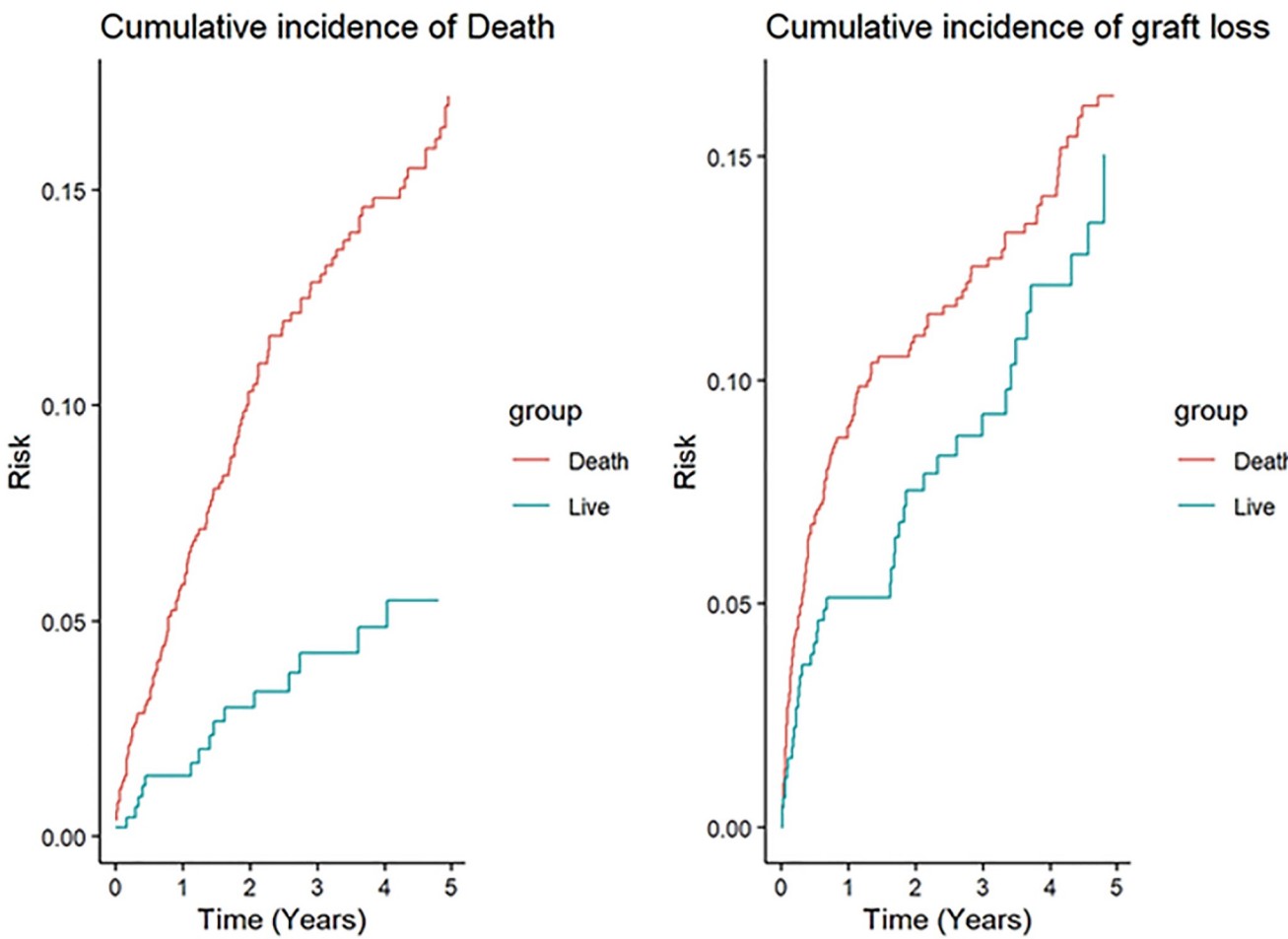

**Fig 2. Log Rank of cumulative incidence of risk of death by type of transplant, and Log Rank of cumulative incidence of risk graft loss by type of transplant.**

cellular rejection, serum creatinine at 12 months and number of hospital readmissions. The covariates that had a significant impact on death were recipient age, diabetes mellitus, type of donor, expanded criteria donor, CMV disease, cold ischemia time, coronary stent, and number of hospital readmissions.

Table 2 provides the $_{sd}$HR of risk factors estimated by using the final multivariate Fine and Gray model. The risk of graft failure was noted to increase in the presence of nephropathy due BK virus, higher rates of serum creatinine at 12 months post transplantation, and greater number of kidney biopsies. Significant risk factors associated with cumulative incidence of death were recipient age, deceased donor, CMV disease, coronary stent, and acute cellular rejection.

Variable selection to build the final model for graft loss was performed comparing three methods:

4. Full model: contains all the predictors selected in previous analysis and no variable selection was done. 2. Backward selection based on the AIC. 3. Backward selection based on the BIC.

The performance and prediction error of the three models were evaluated using Bootstrap cross-validation, showing similar results for the AIC and BIC models. The C-index for the full model was 0.57, for the AIC model was 0.6 and, for the BIC model was 0.6.

**Table 2. Factors associated with graft loss using death as a competing risk in kidney transplant recipients in a final Fine and Gray model.**

| Characteristic | Graft loss outcome | | p-value | Death outcome | | p-value |
|---|---|---|---|---|---|---|
| | sdHR | CI 95% | | sdHR | CI 95% | |
| BKV nephropathy | 4.43 | 2.02–9.72 | <0.001 | - | - | - |
| Serum creatinine at 12 months | 1.76 | 1.55–2.00 | <0.001 | - | - | - |
| Number of renal allograft biopsies | 1.45 | 1.28–1.64 | <0.001 | - | - | - |
| Recipient age (years) | - | - | - | 1.039 | 1.02–1.05 | <0.001 |
| Living donor (Vs deceased) | - | - | - | 0.386 | 0.21–0.68 | <0.001 |
| CMV Disease | - | - | - | 2.459 | 1.46–4.11 | <0.001 |
| Coronary stent | - | - | - | 3.032 | 0.99–9.23 | 0.05 |
| Acute cellular rejection, n (%) | - | - | - | 1.336 | 0.93–1.90 | 0.11 |

sdHR: subhazard distribution; CI: Confidence interval

The same process for variable selection and performance assessment was performed for death model. Similar results were obtained for the AIC and BIC models. The C-index for the full model was 0.78, for the AIC model was 0.72 and, for the BIC model was 0.72. Fig 3 provides the prediction errors and calibration plot of the final Fine and Gray model for graft loss. Fig 4 provides the prediction errors and calibration plot of the final Fine and Gray model for death.

## Discussion

Kidney transplantation is the best therapy available for most patients with end- stage kidney disease [21]. We developed two predictive models of risk of graft loss and risk of death in kidney transplant patients. Risk prediction models are useful for identifying kidney recipients at high risk of graft failure, and optimize clinical care, decision-making and resource allocation; is a challenging issue in kidney transplantation [2]. Our objective was characterized Latin-American cohort of kidney transplant recipients with long term follow-up and to predict the risk factors for graft loss in the presence of mortality as a competing risk event. We were able to identify 3 significant risk factors for graft loss and 5 significant risk factors for death. This contributes to have a better understanding of risk factors for graft loss in a Latin-American population.

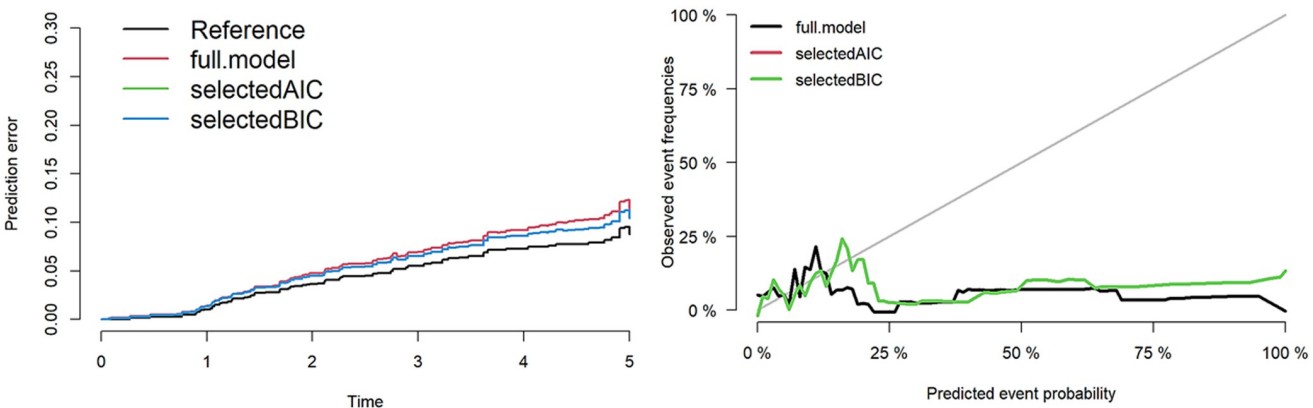

**Fig 3. Prediction errors and calibration plot of the final Fine and Gray model for graft loss.**

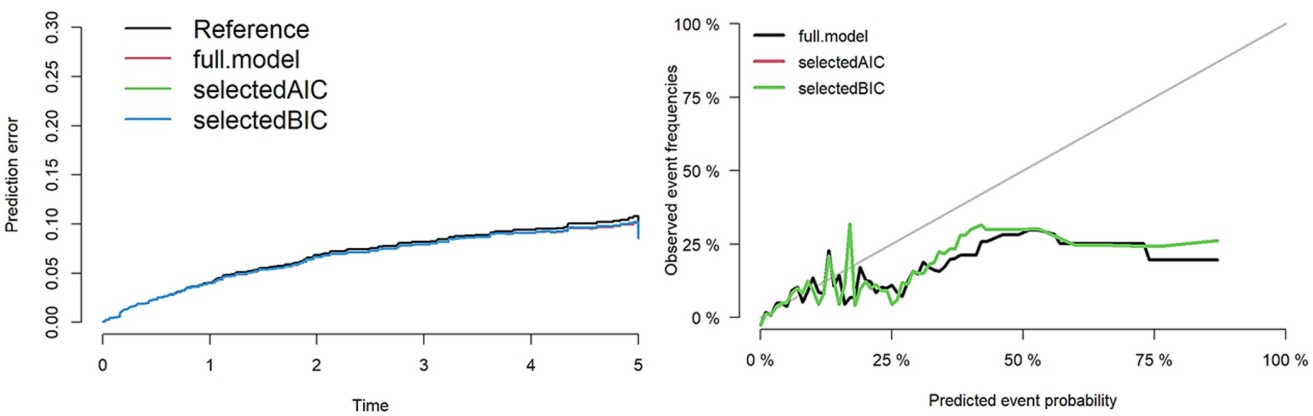

**Fig 4. Prediction errors and calibration plot of the final Fine and Gray model for death.**

Graft survival is one of the most critical concerns in kidney transplant recipients, and our ability to accurately monitor the cumulative incidence of graft loss its importance. Risk prediction models are useful for identifying kidney recipients at high risk of graft failure, thus optimizing clinical care. Therefore, using competing risks methods that provide more accurate estimates, we sought to identify risk factors leading to graft loss, considering death as a competing risk in kidney transplant recipients [4].

Particularly, one study highlights the advantage of using CRA method assessing both the probabilities of death and graft loss.

## Risk factors for graft loss

Late failure of kidney transplants remains an important clinical problem [2, 22]. Renal allograft loss is multifactorial [23]. In the United States, 5469 kidney transplants developed end-stage kidney failure in 2008 (data provided by Jon Snyder from USRDS), making kidney transplant failure the fourth leading cause of end-stage renal disease. The reasons for failure are not well understood. Some have postulated that late deterioration reflects dysregulated fibrosis, drug toxicity or progressive "chronic allograft nephropathy" [24–26]. In our study we found that BKV nephropathy, serum creatinine at 12 months and increased number of renal allograft biopsies were significant risk factors for graft loss. Sellarés et al., attributed causes of graft failure in the biopsy-for-cause population to antibody-mediated rejection (ABMR), probable ABMR or mixed rejection, with nonadherence recorded in nearly half. There was evidence of ABMR in 18 of 19 nonadherent patients who failed. There were also three groups of nonrejection causes of failures: glomerulonephritis, BKV nephropathy and failure in the context of an intercurrent illness. The results emphasize the burden of ABMR and mixed rejection and its interaction with nonadherence in observed failures, making these key targets for further progress. The results also illustrate the range of clinical courses leading to failure and the sometimes-complex relationships to the indication biopsy findings [22].

On the other hand, renal allograft biopsy (RAB) is still the best approach to diagnose renal transplant complications [27]. We found that kidney recipients with more significant requirements to perform RAB had greater risk of graft loss. According to our guidelines, biopsy was performed on those patients with increase of serum creatinine by >20% from baseline, generally when acute or chronic renal allograft rejection is suspected, antibody-mediated rejection, polyoma virus nephropathy, glomerular diseases, atrophy u other. Thus, greater number of RAB may be associated with renal allograft disfunction or detection of other lesions that may

influence graft loss [27–29]. However, as a limitation, we do not have an electronic database of all the histological findings of dysregulated fibrosis and C4d of the renal biopsies of our patients. Of those factors related to graft loss, there is a high magnitude of association with BKV nephropathy [30, 31]. Previous studies have associated the BKV nephropathy with premature loss of kidney function [32–35], graft loss and alteration of renal histology [21–23, 36]. The reactivation of the virus may occur with the use of immunosuppression. Polyomavirus BK virus reactivation in kidney transplant recipients can lead to BK polyoma virus-associated nephropathy (BKPyVAN), which is associated with graft dysfunction in >90% and graft loss in over 50% of the affected individuals [37].

Our results also showed an association with higher serum creatinine level at 12 months. This factor has been widely described as predictor of graft loss [3, 38–45].

The identification of risk factors for graft loss in the long term has been provided by several studies, however, there is substantial variability in data collection, the methods used for model development and included predictors [2]. Among others, the most described predictors are: chronic dysfunction [38, 42, 46], episodes of acute rejection [3, 38–41], death with functional graft [38, 46], glomerulonephritis [38], donor age [47], hypertension [47, 48], diabetes [41, 47], type of immunosuppression [47], delayed graft function [47], recipient age [3], race [3], albumin [3], proteinuria [3, 42, 47], low-density lipoprotein (LDL) cholesterol levels [48] and higher BMI [49]. However, some of them included in the analysis but that finally were not significant.

## Risk factors for death

Identification and quantification of the relevant factors for death can improve patients' individual risk assessment and decision-making [50]. In this study we confirm risk factors for death like recipient age, deceased donor, CMV disease, CMV disease, coronary stent, acute cellular rejection. Our findings show that older recipients are more likely to die, which is consistent with several published studies that report youngest age groups demonstrated a clear trend toward lower mortality compared with those ≥60–65 years [50–53]. However, it must be said that long term patient survival in the elderly has been shown to be significantly better in transplant patients compared with remaining on the waitlist [54–57]. Similarly to what happens with large series (Collaborative Transplant Study [58] and UNOS Register [59]), it is observed that living-donor kidney transplantation provides better outcomes than deceased-donor transplantation. Besides, it is associated with shorter transplant waiting list period and better early outcome [60]. We have found that CMV disease represents a risk of death in our population. This is one of the most important infectious complications in transplant recipient leading to significant morbidity and mortality [61]. Various direct and indirect detrimental effects occur because of CMV infection on patients and grafts. Indirect effects may include rejection, immunosuppression resulting in infections by other microorganisms, graft dysfunction, and poor survival of the kidney graft [62].

Cardiovascular disease (CVD) is frequent after kidney transplantation, is a major cause of morbidity and of death with functioning graft in recipients [63, 64]. We found as a risk factor in the model that coronary disease, specifically coronary stent placement, as a risk factor for death. OPTN/SRTR 2017 Annual Data Report: Kidney, Death with a functioning graft is the leading cause of graft loss in kidney transplant recipients, and a major cause of death is cardiovascular disease, accounting for about one third of known causes [65]. Another of the factors related in the model with the death of kidney transplant patients that we found was the presence of Acute Rejection (AR). Clayton et al., proposed that AR and its treatment may directly or indirectly affect longer-term outcomes for kidney transplant recipients, they found AR was also associated with death with a functioning graft (HR, 1.22; 95% CI, 1.08 to 1.36), and with death due to cardiovascular disease (HR, 1.30; 95% CI, 1.11 to 1.53) and concluded AR is

associated with increased risks of longer-term graft failure and death, particularly death from cardiovascular disease and cancer. The results suggest AR remains an important short-term outcome to monitor in kidney transplantation [66].

Previous studies have attempted to identify and integrate risk factors for death into predictive models, including the pre-transplant variables gender, race, body mass index (BMI), time on dialysis, cause of end-stage renal disease, panel reactive antibodies, HLA mismatches, comorbidities such as diabetes, and heart failure, and donor age. In some models, the post-transplant factors Delayed graft function (DGF), and graft function were included [50], however, in our study population these were not significant. We think that in the case of diabetes the sample size was not sufficient.

## Strengths and limitations

Unlike most previous studies, the main strength of this study is that our analysis includes a competing risk model. Many papers have pointed out the important issue of competing events in kidney transplantation [2, 4, 5, 7]. This method allowed us to determine graft loss risk factors differentiating those who increase recipient mortality. We believe that this integral view is best suited to a rational and patient-centered risk assessment.

Further, our cohort is the largest reporting risk factors for graft loss and death by a unique transplant program in Colombia and contributes to have a better understanding of Latin-American population as most of previous studies have been reported by transplant programs that treat mainly Caucasian patients. Other strengths include consistent data collection with a high degree of completeness and several variables.

Potential limitations attendant with the nature of data collection. The retrospective nature of our study prohibited adjusting for unmeasured confounding factors that may explain the association between independent factors and adverse graft outcomes. Besides, donor age was no considered in our analysis due to no available information.

On the other hand, variable selection with backward regression is not ideal. A fundamental problem with stepwise regression is that some real explanatory variables that have causal effects on the dependent variable may happen to not be statistically significant, while nuisance variables may be coincidentally significant. As a result, the model may fit the data well in-sample but do poorly out-of-sample. Unfortunately, penalized methods for Fine-Gray models have some limitations and the output from the crrp () function does not include convenient parameters such as a p-value. In addition, this package has not been maintained since its first commit in 2015.

We did not perform external validation, and this could be useful to assess the generalizability to other similar populations.

## Conclusion

In summary, we found that stroke, BKV nephropathy, serum creatinine at 12 months and increased number of renal allograft biopsies were significant risk factors for graft loss. On the other hand, recipient age, acute cellular rejection, CMV disease were risk factors for death, and recipients with living donor had better survival compared to deceased-donor transplant and coronary stent. This contributes to have a better understanding of Latin-American population. However, the predictive performance of the models was mild.

## Supporting information

**S1 File.**
(DOCX)

## Acknowledgments

We are grateful with Colombiana de Trasplantes to make this study possible.

## Author Contributions

**Conceptualization:** Jessica Pinto-Ramirez, Andrea Garcia-Lopez, Sergio Salcedo-Herrera, Gilma Hernandez-Herrera, Fernando Giron-Luque.

**Data curation:** Jessica Pinto-Ramirez, Andrea Garcia-Lopez, Sergio Salcedo-Herrera, Nasly Patino-Jaramillo, Juan Garcia-Lopez, Jefferson Barbosa-Salinas, Sergio Riveros-Enriquez.

**Formal analysis:** Jessica Pinto-Ramirez, Andrea Garcia-Lopez, Nasly Patino-Jaramillo, Juan Garcia-Lopez, Jefferson Barbosa-Salinas, Sergio Riveros-Enriquez, Gilma Hernandez-Herrera.

**Investigation:** Andrea Garcia-Lopez, Gilma Hernandez-Herrera, Fernando Giron-Luque.

**Methodology:** Jessica Pinto-Ramirez, Andrea Garcia-Lopez, Sergio Salcedo-Herrera, Juan Garcia-Lopez, Jefferson Barbosa-Salinas, Sergio Riveros-Enriquez, Gilma Hernandez-Herrera.

**Project administration:** Nasly Patino-Jaramillo, Fernando Giron-Luque.

**Supervision:** Jessica Pinto-Ramirez, Sergio Salcedo-Herrera, Gilma Hernandez-Herrera, Fernando Giron-Luque.

**Validation:** Jessica Pinto-Ramirez, Andrea Garcia-Lopez, Gilma Hernandez-Herrera, Fernando Giron-Luque.

**Visualization:** Jessica Pinto-Ramirez.

**Writing – original draft:** Jessica Pinto-Ramirez, Andrea Garcia-Lopez, Nasly Patino-Jaramillo, Juan Garcia-Lopez.

**Writing – review & editing:** Jessica Pinto-Ramirez, Andrea Garcia-Lopez, Nasly Patino-Jaramillo.

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
