## [Decision Letter · Decision Letter 0]

28 Sep 2021

PONE-D-21-25029Risk factors for graft loss and death among kidney transplant recipients: A competing risk analysisPLOS ONE

Dear Dr. García-Lopez,

Thank you for submitting your manuscript to PLOS ONE. After careful consideration, we feel that it has merit but does not fully meet PLOS ONE’s publication criteria as it currently stands. Therefore, we invite you to submit a revised version of the manuscript that addresses the points raised during the review process.

ACADEMIC EDITOR: Please correct the manuscript according to Reviewers' comments. Please pay special attention to Reviewer 2' comments on statistical analysis.==============================

We look forward to receiving your revised manuscript.

Kind regards,

Justyna Gołębiewska

Academic Editor

PLOS ONE

Journal Requirements:

2. We note that your study involved tissue/organ transplantation. Please provide the following information regarding tissue/organ donors for transplantation cases analyzed in your study.

1. Please state in your response letter and ethics statement whether the transplant cases for this study involved any vulnerable populations; for example, tissue/organs from prisoners, subjects with reduced mental capacity due to illness or age, or minors.

- If a vulnerable population was used, please describe the population, justify the decision to use tissue/organ donations from this group, and clearly describe what measures were taken in the informed consent procedure to assure protection of the vulnerable group and avoid coercion. 

- If a vulnerable population was not used, please state in your ethics statement, “None of the transplant donors was from a vulnerable population and all donors or next of kin provided written informed consent that was freely given.”

2. In the Methods, please provide detailed information about the procedure by which informed consent was obtained from organ/tissue donors or their next of kin. In addition, please provide a blank example of the form used to obtain consent from donors, and an English translation if the original is in a different language.

3. Please indicate whether the donors were previously registered as organ donors. If tissues/organs were obtained from deceased donors or cadavers, please provide details as to the donors’ cause(s) of death.

4. Please provide the participant recruitment dates and the period during which transplant procedures were done (as month and year). 

5. Please discuss whether medical costs were covered or other cash payments were provided to the family of the donor. If so, please specify the value of this support (in local currency and equivalent to U.S. dollars)

3. We note that you have stated that you will provide repository information for your data at acceptance. Should your manuscript be accepted for publication, we will hold it until you provide the relevant accession numbers or DOIs necessary to access your data. If you wish to make changes to your Data Availability statement, please describe these changes in your cover letter and we will update your Data Availability statement to reflect the information you provide

Reviewers' comments:

Reviewer's Responses to Questions

**Comments to the Author**

1. Is the manuscript technically sound, and do the data support the conclusions?

Reviewer #1: Partly

Reviewer #2: Partly

Reviewer #3: Yes

2. Has the statistical analysis been performed appropriately and rigorously? 

Reviewer #1: Yes

Reviewer #2: No

Reviewer #3: Yes

3. Have the authors made all data underlying the findings in their manuscript fully available?

Reviewer #1: Yes

Reviewer #2: No

Reviewer #3: No

4. Is the manuscript presented in an intelligible fashion and written in standard English?

Reviewer #1: Yes

Reviewer #2: No

Reviewer #3: Yes

5. Review Comments to the Author

Reviewer #1: Here are my comments/questions to the authors:

Abstract: the conclusion does not reflect the results provided in that section which may confuse a reader that has not yet read the paper.

Introduction:

First paragraph providing details on CKD epidemiology not needed.

Line 78-79: most helpful to say: death is competing event to graft loss because a patient may die before losing the graft, as such no opportunity for graft loss in that case.

Please clarify the difference between CFI and CRF: authors seem to use the terms interchangeably, best to limit to one term to avoid confusion.

Methods

152: AIC and BIC methods: please spell out the first time mentioned (I did not see that).

Results

Tables 1 and 2: would be helpful to either put the p values or indicate where the p values are significant

Table 2: post-op complications: the ureteral leak statistic under graft loss (section of no graft loss) is not clear where it came from: 170 (77.5%) is that the percentage of total of complications: does not fit with the remainder of statistics. Please clarify if typo.

Table 2: any data on histology such as TG or IFTA and effect on graft survival?

Tables 3 and 4:

I am surprised that stroke was found to be statistically significant association with graft loss independent of other factors: how do you explain this result? It is a reflection of uncontrolled cardiovascular risk factors such as HTN and HLD?

I am sceptical of the data related to number of biopsies as independent association with graft loss: unless the rate of biopsy complications at the centers studied is very high, the number of biopsies itself should not be a risk factor for graft loss, rather a reflection of some persistent allograft dysfunction requiring multiple biopsies (indication bias). This would typically be either high creatinine (authors included the one year creatinine but might be useful to look at earlier time points such as 3 or 6 months and later such as 2 years) and proteinuria (which is not included here and would be very useful to include in the analysis).

Expanded criteria donor is an older term best replaced with the KDPI if available.

I am surprised that DM did not make it as an independent association with graft loss and seems to have been supplanted by obesity (although the actual HR in the univariate analysis for DM were higher than for obesity). This is a very interesting result which seems to suggest that obesity rather than diabetes is responsible for the excess mortality. Please clarify: 1) history of DM: is that only pre-transplant DM or was new onset post-transplant DM included?. 2) what is the proportion of DM type 1 vs type 2?

Discussion

220: please include reference 9 in that sentence as well.

You have the same reference listed as 9 and 19: please consolidate.

Line 243: I would argue that there is a significant amount of data that links BK nephropathy with graft loss and unfavorable histology including the following that would be important to acknowledge:

Gago et al: Kidney allograft inflammation and fibrosis, causes and consequences. PMID: 22221836

Sellares et al: Understanding the causes of kidney transplant failure: the dominant role of antibody-mediated rejection and nonadherence. PMID: 22081892

El-Zoghby et al: Identifying specific causes of kidney allograft loss. PMID: 19191769

Naesens et al: The histology of kidney transplant failure: a long-term follow-up study. PMID: 25243513

Paragraph 260-266: I am not clear as to the point of this paragraph. Please clarify how does that link to your analysis and how your analysis is better and risk factors that you included are better than what you listed from the existing literature. I do agree as pointed above that proteinuria is an important risk factor that is not included in your analysis.

You also allude to unfavorable histology as possible explanation to the number of biopsies being a risk factor and stroke: as I asked above, would be interesting to have that data analyzed in your cohort.

In the paragraph of risk factors for death, it is important to address why DM did not remain an independent risk factor of death (contrary to other studies) and give possible hypotheses for that observation.

Reviewer #2: Since the KM model does not account for competing risk, the KM model should not be used at all. As the title clearly state “a competing risk analysis”, the KM model should be omitted. The author may try different competing risk models though.

The conclusion is not accurate and did not focus on the primary aim of this paper which is to identify risk factors. Even though the results of KM and competing risk analysis are different, you cannot claim that KM overestimates the incident based on a specific data. You need to perform more rigorous methodology analysis to compare and validate the two models.

A propensity score matching should be performed.

Were the final models in Table 4 selected by p value or AIC/BIC? Please clarify.

Minor concerns:

Use “univariable/multivariable” instead of “univariate/multivariate”.

Cite the reference to the competing risk analysis.

Add p values to Table 1 and 2.

Table 3 footnote: “IC” should be “CI”.

The “Model performance” section in the results seems to be more “methods” not “results”.

Reviewer #3: I thank the authors and editors for the opportunity to review this interesting manuscript.

Summary

Using a 10-year long retrospective cohort of first-time kidney transplant from Columbia, Pinto-Ramirez et al. applied competing risks analyses and used forward stepwise predictor selection to identify risk factors for graft failure and patient survival. Stroke, BKV nephropathy, acute cellular rejection, serum creatinine, and increased number of renal biopsies were significant predictors for graft loss, while PRA II negative was protective. Recipient age, obesity, and cytomegalovirus disease were associated with an increased risk of death, while living donor transplantation was protective.

Major

It is important for the authors to clarify the main message of the manuscript. Is it to (i) educate the readership on competing risks analysis or (ii) respond to the main aim of identifying risk factors for graft failure and patient survival among kidney transplant recipients from Columbia? While application of suitable analytical methods as outlined in the METHODS section is encouraged, the discrepancy between KM and CIF estimates (derived from competing risks analysis) in and of itself, is not novel. Thus, the main contribution of this manuscript relates to the development of prediction models in kidney transplant recipients form Columbia. This could be better emphasized in the INTRODUCTION and DISCUSSION sections of the article.

Consultation of the STROBE and TRIPOD statements on reporting of observational studies and prediction models, respectively, is advisable.

Additional points for consideration regarding predictor variable definition and handling, as well as for prediction model development are outlined below:

ABSTRACT:

- The METHODS section should clarify predictor variable definition and handling, as well as the variable selection process for each model depending on outcome

- The main conclusion should be aligned with the main objective(s) of the manuscript.

INTRODUCTION:

- Kindly review previously published prediction models for similar outcomes in kidney transplant recipients and the important predictors identified from those.

METHODS:

- Please mention protocols for treating rejection

- Please discuss HLA (in)compatibility assessment pre-transplant and how it affects allocation decisions

- Definition, handling, and timing of predictor measurement

o What period was considered for the “number of biopsies” variable definition?

o Why was donor sex not considered among the predictor variables?

- Multiple imputation for handling missing values is advisable

- Modelling technique and predictor selection

o Why were no prespecified variables (based on published literature) included in the prediction model?

o Please discuss how correlation between variables was considered prior to fitting the final prediction model

o Please consider LASSO (least absolute shrinkage and selection operator) for fitting parsimonious prediction models for the study endpoints.

o Given the consideration of post-transplant characteristics (e.g., 12-month creatinine and number of biopsies), was landmark analysis or time-varying analysis considered?

- Please specify the timeframe the models intend to prognosticate for

- Justification is needed for lack of internal validation of the final prediction models

RESULTS:

- Consider organizing variables presented in Table 3 based on recipient, donor, and transplant characteristics. For the latter, also consider timing in reference to transplantation

- Is it “biological sex” or “gender” that is considered among the predictor variables?

- FIGURES 1 to 3

o Please use a similar scale for the y-axis for all figures.

o Please modify Y-axis for Figure 1 to indicate CIF of graft failure and death

o Please clarify if Figure 3 provides CIF and Gray test or K-M estimates and log rank

- The model performance section in its current form is better placed in the METHODS section. Please report on model discrimination (e.g., C-statistic) with confidence intervals in the RESULTS

- For which models were AIC and BIC compared?

- For the final models, please provide equations including intercept and regression coefficients

DISCUSSION:

- The first paragraph should summarize the main novel observations. Currently it reads as the rationale for the study.

- Limitations

o Please discuss limitations of forward stepwise selection for prediction model specification

o Please mention and justify lack of internal validation (as well as external validation)

- Please explain the utility of the final prediction models and how they could be incorporated in clinical care

Minor

- Line 95: “competitive” should be replace with “competing”

- Line 134: “available case analysis” should be replaced with “complete case analysis”

- Define PRA II negative (i.e., no preformed antibodies against class II HLA)

- Does BMI in Table 2 relate to donor or post-transplant recipient BMI?

- Define “Match” (e.g., HLA-A,-B,-DR antigens)

- Which classification was used for rejection diagnoses?

6. PLOS authors have the option to publish the peer review history of their article (what does this mean?). If published, this will include your full peer review and any attached files.

Reviewer #1: No

Reviewer #2: No

Reviewer #3: **Yes: **Ruth Sapir-Pichhadze

---

## [Author Response · Author response to Decision Letter 0]

13 Apr 2022

Bogotá DC, Abril 1th 2022

Dear Doctor, 

Emily Chenette, PhD

Editor-in-Chief

PLOS ONE

Response Letter 

We summarized the corrections of the research manuscript “Risk factors for graft loss and death among kidney transplant recipients: A competing risk analysis”, please see below: 

Journal requirements

1. We ensured our manuscript met PLOS ONE's style requirements. 

2. Ethics statement: None of the transplant donors was from a vulnerable population and all donors or next of kin provided written informed consent that was freely given. 

3. We provided the original and the English version of the “Organ and Tissue Donation Consent Form” from deceased donors.

4. The participant recruitment dates and the period during which transplant procedures were done are provided in the methods section. 

5. In the methods section is specified thar “the total of donor medical costs is covered including organ donor maintenance and procurement with an average of 5000 USD and without any economic contribution to the family donor”. 

6. We provided a repository of information for our data that is available with DOI: 10.17632/3j2m4ftn69.1 

Reviewer 1 

 Abstract: 

- We changed the conclusion 

 Introduction: 

1. CKD epidemiology was removed from the manuscript. 

2. We corrected the paragraph in lines 78-79 about death as a competing event to graft loss.

3. We did not use the Competing Random Forest (CRF) to analyze our data. Instead, we used the Cumulative Incidence Function (CIF).

Methods 

4. Line 152: We spelled out The Akaike information criterion (AIC) and the Bayesian Information Criterion (BIC). 

5. Tables 1 and 2: We added the p values that are significant in Table 1, and we removed further Tables that were all summarized in one table. 

6. Table 2: Table 2 was removed 

7. Table 2: We did not have any TG or IFTA data available to analyze the effect on graft survival. Table 2 was removed.

8. Tables 3 and 4: Our center does not perform protocol biopsies. We perform kidney biopsies only with clinical indications. Thus, this could be the reason for the correlation between the number of biopsies and graft loss (indication bias). Tables 3 and 4 were summarized in Table 2.

9. We do not have KDPI available for donors in our country. 

10. 1)We included only pretransplant DM, 2) We did not have available data about the proportion of DM type 1 and 2. 

 Discussion:

11. Line 220: We added reference 9 and consolidate it. 

12. Line 243: We included the articles suggested about BK nephropathy and graft loss.

13. Paragraph 260-266: in the methods section we specified how were collected the potential risk factors for graft loss, we included those published from the existing literature and also those available in our data. Those predictors not included are recognized as a limitation. 

14. Why DM did not remain an independent risk factor of death (contrary to other studies)? We think that in the case of diabetes the sample size was not sufficient

Reviewer 2

1. Since the KM model does not account for competing risk, we removed the KM model.

2. We changed the conclusion and focused it on the aim of this paper which is to identify risk factors. 

3. Propensity scores are used to balance observed covariates between subjects from the study groups to mimic the situation of a randomized trial (Joffe & Rosenbaum, 1999) and can be used for matching, stratification, or in a regression model as a covariate or weight (Rubin, 1997; D’Agostino, 1998). Because propensity scores are used to address potential confounding by indication, they would not be expected to improve pure prediction, which is not concerned with specific coefficient estimation. Additionally, propensity scores are estimated from regressions that comprise the same covariates included in the traditional prediction models, and only those covariates, thus it would seem mathematically impossible for the propensity scores to add anything – they are simply functions of the same variables already included in the traditional models. Despite this argument, requests for the addition of propensity scores to pure prediction models persist https://www.ncbi.nlm.nih.gov/pmc/articles/PMC3740143/

4. We specified how the final models were selected in the methods section.

5. We used “univariable/multivariable” instead of “univariate/multivariate”.

6. We cited the reference to the competing risk analysis.

7. We Added p values to Table 1 and 2. (now summarized in table 1)

8. Table 3 footnote: “IC” should be “CI”. (now in table 2)

9. We removed The “Model performance” section from the results 

Reviewer 3

Major 

1. We emphasized the development of prediction models in the INTRODUCTION and DISCUSSION sections of the article. 

2. We consulted and used STROBE and TRIPOD statements

ABSTRACT: 

3. We clarified the predictor variable definition and handling, as well as the variable selection process for each model depending on outcome in the methods section 

4. The main conclusion was improved to be aligned with the main objective(s) of the manuscript. 

INTRODUCTION: 

5. We reviewed previously published prediction models for similar outcomes in kidney transplant recipients and the important predictors identified from those. 

METHODS: 

6. We mentioned protocols for treating rejection 

7. We discussed HLA (in)compatibility assessment pre-transplant and how it affects allocation decisions.

8. We provided the list of predictors and variables in a supplementary material. We also clarified predictors in the method section

9. Multiple imputation for handling missing values was no performed as there were few missing values

10. We specified the Modelling technique and predictor selection

11. We considered LASSO for fitting parsimonious prediction models. Unfortunately, for the package crrp:::crrp (this package is useful for penalized Fine-Gray models, using LASSO, SCAD, MCP, and their group versions) there is no predict function, and the output from the crrp() function does not include convenient parameters such as a P value. There are standard errors, so a P value can be manually calculated. Creating a tidy wrapper may be difficult. In addition, this package has not been maintained since its first commit in 2015.

 12. Analysis of time-dependent covariates has some limitations in both the survival analysis and the competing risks setting. When different causes are acting simultaneously, the main interest is in estimating quantities such as cause-specific hazards, cumulative incidences or marginal survival probabilities. If random (internal) time-dependent covariates are to be included in the modeling process, then it is still possible to estimate cause-specific hazards but prediction of the cumulative incidences and survival probabilities based on these is no longer feasible (Andersenet al., 1993, Chapter VII; Kalbfleisch and Prentice, 2002, Chapter 6). This limitation is also encountered in the regression approach for cumulative incidences (or ‘‘subdistribution hazards’’) (Fine and Gray, 1999) and has been further discussed by Latouche, Porcher, and Chevret (2005) and Latoucheet al.(2007) and tackled by Beyersmann and Schumacher (2008)

13. We specified the timeframe the model’s intent to prognosticate for in the methods section (5 years postrasplant) 

14. Internal validation of the models was performed.

RESULTS: 

15. Table 3 was removed.

16. Biological sex was considered among the predictor variables, not gender. 

17. Figure 3, is now figure 2. We change the scale of figures. 

18. We indicated that figure 3 (now figure 2) shows CIF of graft failure and death

19. The model performance section is explained in the methods section.

20. The model discrimination and calibration were reported 

21. Characteristics of modelling are provided in a supplementary material

DISCUSSION: 

- We summarized the main novel observations in the first paragraph

-We discussed the limitations of stepwise regression selection. 

- We performed internal validation using resampling techniques.

Minor 

1. Line 95: We replaced “competitive” with “competing”. 

2. Line 134: We replaced “available case analysis” with “complete case analysis”.

3. We defined PRA II negative in Table 1. 

4. We corrected the observation about BMI in Table 2 which is related to the post-transplant recipient BMI (Now summarized in Table 1). 

5. We gave a definition of a match. 

6. We added the Banff classification for rejection diagnoses.

---

## [Decision Letter · Decision Letter 1]

2 Jun 2022

Risk factors for graft loss and death among kidney transplant recipients: A competing risk analysis

PONE-D-21-25029R1

Dear Dr. Pinto-Ramirez,

We’re pleased to inform you that your manuscript has been judged scientifically suitable for publication and will be formally accepted for publication once it meets all outstanding technical requirements.

Kind regards,

Justyna Gołębiewska

Academic Editor

PLOS ONE

Additional Editor Comments (optional):

Reviewers' comments:

Reviewer's Responses to Questions

**Comments to the Author**

1. If the authors have adequately addressed your comments raised in a previous round of review and you feel that this manuscript is now acceptable for publication, you may indicate that here to bypass the “Comments to the Author” section, enter your conflict of interest statement in the “Confidential to Editor” section, and submit your "Accept" recommendation.

Reviewer #2: All comments have been addressed

2. Is the manuscript technically sound, and do the data support the conclusions?

Reviewer #2: (No Response)

3. Has the statistical analysis been performed appropriately and rigorously? 

Reviewer #2: (No Response)

4. Have the authors made all data underlying the findings in their manuscript fully available?

Reviewer #2: (No Response)

5. Is the manuscript presented in an intelligible fashion and written in standard English?

Reviewer #2: (No Response)

6. Review Comments to the Author

Reviewer #2: (No Response)

7. PLOS authors have the option to publish the peer review history of their article (what does this mean?). If published, this will include your full peer review and any attached files.

Reviewer #2: No

---

## [Editor Report · Acceptance letter]

6 Jul 2022

PONE-D-21-25029R1 

Risk factors for graft loss and death among kidney transplant recipients: A competing risk analysis 

Dear Dr. Pinto-Ramirez:

I'm pleased to inform you that your manuscript has been deemed suitable for publication in PLOS ONE. Congratulations! Your manuscript is now with our production department. 

Kind regards, 

on behalf of

Dr. Justyna Gołębiewska 

Academic Editor

PLOS ONE